# Anti-PD-1/PD-L1 Based Combination Immunotherapy to Boost Antigen-Specific CD8^+^ T Cell Response in Hepatocellular Carcinoma

**DOI:** 10.3390/cancers13081922

**Published:** 2021-04-16

**Authors:** Julia Peña-Asensio, Henar Calvo, Miguel Torralba, Joaquín Miquel, Eduardo Sanz-de-Villalobos, Juan-Ramón Larrubia

**Affiliations:** 1Translational Hepatology Unit, Guadalajara University Hospital, 19002 Gudalajara, Spain; julia.pena@edu.uah.es (J.P.-A.); hcalvo@sescam.jccm.es (H.C.); miguel.torralba@uah.es (M.T.); jmiquelp@sescam.jccm.es (J.M.); eduardos@sescam.jccm.es (E.S.-d.-V.); 2Department of Biology of Systems, University of Alcalá, 28871 Alcalá de Henares, Spain; 3Section of Gastroenterology & Hepatology, Guadalajara University Hospital, 19002 Guadalajara, Spain; 4Service of Internal Medicine, Guadalajara University Hospital, 19002 Guadalajara, Spain; 5Department of Medicine & Medical Specialties, University of Alcalá, 28871 Alcalá de Henares, Spain

**Keywords:** hepatocellular carcinoma, immunotherapy, PD-1, PD-L1, immune check-point inhibitor, combination therapy, CD8 T cell response

## Abstract

**Simple Summary:**

The cytotoxic T cell response against hepatocellular carcinoma antigens is exhausted and fails in its task of deleting tumoral cells. These cells are featured by the expression of negative immune checkpoints that can be modulated to restore T cell function. The blockade of the PD-1/PD-L1 pathway has shown promising results in rescuing hepatocellular carcinoma-specific CD8 T cells but only a reduced group of cases is sensitive to this treatment and the effect is usually temporary. Therefore, new anti-PD-1 based combinatory strategies are underway to increase the response by adding the effect of blocking neo-angiogenesis and other negative immune checkpoints, boosting positive immune checkpoints, blocking suppressive cytokines, or inducing the expression of tumoral neoantigens. The restoration of T cell responses with these anti-PD-1 based combinatory therapies will change the outcome of advanced hepatocellular carcinoma.

**Abstract:**

Thirty to fifty percent of hepatocellular carcinomas (HCC) display an immune class genetic signature. In this type of tumor, HCC-specific CD8 T cells carry out a key role in HCC control. Those potential reactive HCC-specific CD8 T cells recognize either HCC immunogenic neoantigens or aberrantly expressed host’s antigens, but they become progressively exhausted or deleted. These cells express the negative immunoregulatory checkpoint programmed cell death protein 1 (PD-1) which impairs T cell receptor signaling by blocking the CD28 positive co-stimulatory signal. The pool of CD8 cells sensitive to anti-PD-1/PD-L1 treatment is the PD-1dim memory-like precursor pool that gives rise to the effector subset involved in HCC control. Due to the epigenetic imprints that are transmitted to the next generation, the effect of PD-1 blockade is transient, and repeated treatments lead to tumor resistance. During long-lasting disease, besides the TCR signaling impairment, T cells develop other failures that should be also set-up to increase T cell reactivity. Therefore, several PD-1 blockade-based combinatory therapies are currently under investigation such as adding antiangiogenics, anti-TGFβ1, blockade of other negative immune checkpoints, or increasing HCC antigen presentation. The effect of these combinations on CD8^+^ T cells is discussed in this review.

## 1. Introduction

Hepatocellular carcinoma (HCC) is the second leading cause of cancer-related death worldwide and has an incidence of approximately 850,000 new cases per year. HCC accounts for approximately 90% of malignant liver tumors and is one of the most aggressive types of cancer, with few and unsatisfactory therapeutic options [1]. Since current therapies are limited and recurrence of HCC is common in patients treated with the standard therapeutic arsenal, immunotherapy presents itself as a promising possible response to the real need for new treatments [2]. The new emergence of treatments based on the modulation of inhibitory immune checkpoint (IC) opens new opportunities for the control of hepatocarcinoma [3,4,5]. Blockade of the programmed cell death protein 1 (PD-1)/PD-ligand(L)1 pathway is the paradigm of this type of treatment to counteract T cell response exhaustion [6,7], but we still do not have predictive variables to forecast a favorable response, and in many cases the initial response disappears [8,9]. The HCC-specific CD8 T cell response can be essential in HCC control due to its ability to recognize tumor cells and destroy them by cytolytic and no cytolytic mechanisms [10]. Nevertheless, these cells become exhausted during cancer progression [11,12] but can be temporally restored by immune checkpoint inhibitors (ICI). To enhance and extend the effect of this approach, the synergy of this therapy with other strategies acting at different levels (blocking other negative ICs, triggering positive co-stimulatory pathways, blocking vascular endothelial growth factor (VEGF) pathway, reprogramming mitochondrial metabolism, or adding loco-regional therapies) is being explored [13]. This work revises the types of HCC potentially sensitive to immunotherapy, the role of PD-1 expressing CD8 T cells in HCC treatment, the target population for CD8 T cell immunotherapy and the effect of PD-1 based treatment combinations on the restauration of HCC-specific cytotoxic T cell response.

## 2. Types of HCC According to Potential Sensitivity to PD-1 Based Immunotherapy

HCC is an inflammation-driven disease induced by chronic viral hepatitis (hepatitis B (HBV) and C (HCV) viruses) and non-viral “sterile” inflammation (alcohol abuse, non-alcoholic steatohepatitis (NASH) and other rare metabolic disorders, such as hemochromatosis or α-1 antitrypsin deficiency) [14]. Despite the different etiologies, the immune compositions are similar between HBV/HCV infected and virus negative tumors, apart from activated M2 macrophages that are observed mainly in virus-related HCC [15]. Nevertheless, the HCC-specific CD8 T cell response could be weaker in NASH-related HCC than in HCC associated with hepatitis virus infection [16], which could impact on immunotherapeutic strategies.

Previous extensive immunogenomic analysis of different type of tumors have shown diverse immune signatures. Based on these clusters, HCC can be classified according to the features of the induced immune response. Those non-immunogenic tumors are characterized by lymphocyte depletion or being immunologically quiet. Immunotolerant cancers are featured by the phenotypes “wound healing”, inflammatory and tumor growth factor (TGF)-β1 dominant. The immunogenic tumors comprise those cases that are interferon (IFN)-γ dominant [17]. The lymphocyte depleted cluster includes around 40–55% of HCC cases and are featured by M2 macrophage infiltration, moderate genomic heterogenicity, and high level of mutations in the catenin–β1 gene (*CTNNB1*). The “wound healing” HCCs are around 5–15% of cases and have a balance between macrophages and lymphocytes, high genomic heterogenicity, and mutations in the tumor suppressor gene TP53. The IFNγ dominant HCCs comprise around 10–20% of cases and are rich in CD8^+^ T cells and M1 macrophages with high PD-1 expression and T cell receptor (TCR) diversity. The HCC inflammatory type includes around 15–30% of cases and is featured by T helper (Th)17 infiltration, high Th1 response, low genomic heterogenicity, and high PD-L1 expression. Finally, the TGF-β1 dominant and the immunologically quiet clusters are poorly represented in HCC [17,18]. These different phenotypes should be considered to adapt the strategies to boost the immune response, as will be discussed at the end of this review (Figure 1). The detection of immunogenic HCC could be a key step to select those cases prone to respond to ICI. In those non-immune cases, strategies to induce a CD8^+^ T cell tumoral infiltration should be taken into consideration.

The liver is an immunotolerant environment due to its need to deal with the exposure to many gut-derived exoantigens [19]. This special liver characteristic could facilitate the immune evasion in HCC, inducing either a defective T cell response or its physical deletion [20], linked to the induction of inhibitory IC such as PD-1 and its ligands [21,22]. Nevertheless, at HCC diagnosis not all tumors show data of complete HCC immune suppression. Trying to stratify the type of tumor that could respond to PD-1/PDL-1 manipulation, researchers have described another classification based on four scenarios according to PD-L1 expression. Type I tumors display tumor infiltrating lymphocytes (TIL) and PD-L1 expression on tumor cells. These features would suggest that an adaptive immune resistance is ongoing. Type II cases neither express PD-L1 nor have TIL, which indicates lack of immune response. Type III tumors express PD-L1 but without TIL, indicating an intrinsic induction of PD-L1. Finally, type IV tumors are PD-L1 negative but with TIL, suggesting that in these cases other mechanisms are involved in the development of intra-tumoral T cell exhaustion [23]. HCC patients with PD-L1 expression in the tumor have poorer overall survival (OS) than PD-L1 negative cases [24], but this can also be a marker of higher CD8 infiltration and higher survival after treatment [25]. In a recent metanalysis, PD-L1 positive patients had a risk of death higher than the double with respect to PD-L1 negative cases [26]. Around 20% of HCC expresses PD-L1 in more than 1% of tumor cells, and this correlates with a worse progression [27]. Fortunately, in the anti-PD-1 clinical trials with nivolumab and pembrolizumab (ChekMate-40 and KEYNOTE-224), those cases with higher PD-L1 expression had a higher response rate with a longer OS, but some patients with lower PD-L1 expression also responded to this treatment [28,29,30]. This response in negative cases could be due to biopsy sampling error because of the heterogenicity of PD-L1 expression in the HCC, since in resected HCC, PD-L1 detection has been reported in more than 80% of cases [31]. Consequently, this variable tested in a liver biopsy could not be useful to decide which patients should receive treatment to block the PD1/PD-L1 pathway, although it could be used as a prognosis marker. Consequently, high PD-L1 expression could be a marker of worse progression but also a prognosis factor for better treatment response. In any case, the role of PD-L1 expression to plot a therapy based on anti-PD-1 treatment is not clear yet and it must be clarified in future studies.

Another specific analysis of gene expression profiles from tumor, stromal, and immune cells in a big cohort of HCC showed that 25% of patients have a genetic signature related to the induction of an immune response [32]. These cases displayed a high level of PD-1 expression and markers of cytolytic activity. Within this “Immune Class” HCC, two different groups were described; those characterized by an effective response and those in whom there is an exhausted immune response that expressed many genes regulated by TGF-β1 [32]. These immune HCC types could correspond to the IFNγ dominant and the immunotolerant groups previously discussed in this section [17]. These distinct immunologic signatures correlate with different HCC outcomes. Those cases with an active immune response phenotype had a longer OS than those with an exhausted or non-detectable response [32]. The PD-1 up-regulation on this “Immune class” tumors could make them sensitive to ICI.

An important issue in HCC-specific CD8 T cell response induction is the immunogenic capacity of the potential HCC antigens. The T cell response against aberrant expression of host’s antigen, such as glypican-3, NY-ESO-1, MAGE-A1, and MAGE-A3 are not commonly efficiently induced in HCC and subsequently cannot be considered the backbone for immunotherapy [33]. Nevertheless, intrahepatic T cells targeting immunogenic tumoral neo-antigens display an exhausted phenotype that suggests tumor recognition [34]. Therefore, immunotherapeutic rescue of T cells targeting tumor neo-antigens could be the best approach to improve tumor control [35]. HCC has an intermediate level of neo-antigen expression, which could be an advantage for immunotherapeutic strategies. Consequently, the existence of neo-antigens could be a premise for the presence of sensitive HCC-specific CD8 T cells to ICI. Therefore, tumor mutational burden (TMB) could be a predictive biomarker for the efficacy of ICI therapy [36]. Specifically, HCC develops regularly neoantigens although less frequently than other tumors, such as melanoma or lung cancer [37]. Nevertheless, in HCC the frequency of cases with high TMB is low and does not correlate with the rate of predicted neo-antigens [38]. Therefore, the role of TMB as predictor of response to ICI is not clear yet.

According to these data, the analysis of the immune response could be a predictive factor for ICI success. In fact, in those cases with a more immunosuppressive microenvironment such as in virus-associated HCC, a higher PD-1 expression is found, and this could impact on ICI response [39]. Moreover, in the nivolumab trial, those cases with higher T cell reactive liver inflammation correlated with better response and longer OS [32]. We could summarize that a good candidate to restore HCC-specific CD8 T cell response by PD-1/PD-L1 blockade could be considered those tumors with high grade of IFN-γ-secreting TIL, high PD-L1 expression, low M2 macrophage infiltration, and presence of HCC neo-antigens.

## 3. Role of PD-1-Expressing HCC-Specific CD8 T Cell Response in HCC Control

The tumor-associated antigen-specific CD8^+^ T cell response is essential for the control of solid tumors due to their ability to recognize tumor cells and to destroy them [40,41]. Exhausted CD8^+^ T cells are the main subset of TILs that perform anti-tumor effector functions [10,42]. Specifically, in HCC it is possible to find a high number of T cells infiltrating the tumor (Figure 2) and there is a correlation between the density of infiltrating lymphocytes and the prognosis of the disease [43]. Furthermore, after loco-regional treatment of HCC, a longer survival has been observed in those subjects in whom a specific CD8^+^ T response against HCC associated antigens is detected [44]. Therefore, large numbers of CD8^+^ TILs in HCC correlate with improved OS, longer relapse-free survival, and diminished disease progression [10,45,46]. HCC associated antigens comprise a heterogeneous set of autologous proteins that are rendered immunogenic in tumors by mutation [35] or aberrant expression [10,47] (Figure 2). The development of an effector cytotoxic T cell immune response against these antigens may provide a barrier to tumor progression. Nevertheless, most of the T cells targeting autologous aberrantly expressed epitopes are deleted or in a tolerogenic state [33,48], although some studies have shown a positive effect after PD-1 blockade in T cells targeting these kind of antigens [7]. Anyhow, the response to ICI could be more effective in T cells targeting HCC neo-antigens. A previous study has suggested that searching for host’s human leukocyte antigen (HLA) restricted tumoral neoantigens will permit to find T cells that after the appropriate treatment will exert tumor control [35]. HCC will commit mutations that will give rise to new epitopes that will be recognized as exogen antigens by T cells. Unfortunately, these cells will become exhausted during HCC progression, not being able to keep the tumor under control. These exhausted CD8^+^ T cells show a gene signature that is completely different from those of memory and effector T cells [49,50,51]. During the exhaustion process, T cell response progressively loses the effector functions and finally undergoes apoptosis [11,52]. The exhausted CD8 T cells are featured by the expression of different inhibitory IC, such as PD-1, cytotoxic T lymphocyte antigen-4 (CTLA-4), T cell immunoglobulin domain and mucin domain (Tim3), and lymphocyte-activation gene 3 (LAG3) that can be modulated to rescue T cell reactivity [21,53]. The ligands of these IC are expressed by resident liver cells. In fact, PD-L1 is expressed on hepatocytes [22], hepatic stellate cells [54], liver sinusoidal endothelial cell (LSEC) [55], and Kupffer cells [56,57], while PD-L2 expression is restricted to dendritic cells [58]. PD-1 and PD-L1 upregulation promotes CD8^+^ T-cell apoptosis and postoperative recurrence in hepatocellular carcinoma patients [59]. The in-vitro treatment of these T cells with antibodies against these myriad of inhibitory IC increases T cell proliferation and cytokine secretion.

Nevertheless, these exhausted CD8^+^ cells develop epigenetic imprints that steadily maintain the functional impairment, which is transmitted to the progeny, making temporary the effects of immunomodulatory treatments [49]. Even in the context of a PD-1 knockdown HCC-specific CD8 T cell model, although a tumor killing enhancement is initially observed, this effect is limited by compensatory engagement of alternative co-inhibitory and senescence programs upon repetitive antigen stimulation [60]. These epigenetic imprints induce the expression of specific transcription factors, such as the thymocyte selection-associated high mobility group box protein (TOX) that is up-regulated on CD8^+^ T cells in HCC and promotes their exhaustion by regulating endocytic recycling of PD-1 [61]. Additionally, the up-regulation of the ligands of these negative IC is induced in the HCC microenvironment. The regulation of PD-L1 expression could be related with the level of M2 macrophages that are recruited by the tumor cell-intrinsic osteopontin secretion. In HCC animal models, the level of M2 macrophages decreases after osteopintin blockade, which correlates with an increased CD8 effector response to PD-L1 blockade [62]. M2-like macrophages favor an immunosuppressive landscape by depleting CD8 T cells and inducing CD4^+^ T regulatory cells (Tregs) [63]. Additionally, M1 macrophages can induce PD-L1 expression on hepatocarcinoma cells by IL-1β effect [64]. These data suggest an important role of macrophages in modulating CD8 T cell exhaustion in HCC by promoting a immunotolerant environment [65] and by up-regulating themselves both PD-L1 and PD-L2 expression [66]. However, PD-L1 expression on macrophages could also have a positive input because it correlates with CD8 T cell infiltration and increased OS after treatment [67], probably in the case of M1-like macrophage infiltration [17]. Moreover, HCC-specific CD8 T cell itself induces PD-L1 up-regulation on hepatocytes, linked to a subsequent impairment of IFN-γ secretion by T cells [68]. There is a gradient of PD-1 expression on intra-tumoral CD8^+^ cells, which probably defines the progenitor (PD-1^dim^) and effector (PD-1^high^) pools [69]. The PD-1^high^ population can co-express positive co-stimulatory checkpoints, such as 4-1BB, that can be triggered to rescue these cells from exhaustion [70,71]. Nevertheless, to target the PD-1^dim^ subset could be more operative to get a more efficient response to ICI. The PD-1 level of the intrahepatic resident CD8^+^ cells inversely correlates with the expression of the transcription factor T-bet [72], which has been correlated with a late dysfunctional phenotype in the PD-1^high^ pool, since the effector late dysfunctional pool expresses the transcription factor Eomes and loses T-bet expression [73]. On the contrary, PD-1^dim^ expression is a marker of the progenitor pool, which is the subset that provides the proliferative burst after anti-PD-1 treatment [69]. Therefore, T cell restoration should focus on PD-1^dim^ CD8^+^ T cells. In order to improve both the precursor and the effector pools, searching for PD-1/PD-L1 based combinatory therapies, such as modulation of suppressive soluble mediators (interleukin (IL)-10, IL-17, TGF-β1), blocking suppressive cells (Tregs, myeloid derived suppressor cells (MDSC), M2 tumor associated macrophages (TAM)), triggering positive co-stimulation (tumor necrosis factor receptor superfamily member 9, 4-1BB), mitochondrial metabolic reprogramming, impairing neo-angiogenesis, inducing expression of HCC neo-antigens or epigenetic modulation by gamma-chain (γc) cytokines [11,12,14,74] could improve the response to PD-1/PD-L1 blocking monotherapy. Figure 3 summarizes the potential mechanisms involved in HCC-specific CD8 T cell impairment.

## 4. Targeting the Progenitor HCC-Specific CD8 T Cell Pool for Immunomodulation

Unlocking the potential of immunotherapy requires the design of strategies that induce a potent functional memory T cell pool capable of protecting from recurrence by producing a non-exhausted effector subset [75]. The exhausted progenitor pool gives rise to the progeny probably by asymmetric cell division [76,77]. The progeny comprises two sub-populations: effector-like transitory cells and late dysfunctional pool [73]. The effector-like transitory cells are characterized by the expression certain IC, such as PD-1 and Tim-3, and are able to sustain certain degree of tumor control. These cells will differentiate into the late dysfunctional pool, which expresses high levels of multiple negative IC, lacks proliferative capacity, displays low polyfunctionality, but retains some killing capability and expresses the transcription factor Eomes. Due to the high PD-1 expression on these cells, they are not suitable for anti-PD-1 therapy [73]. The progenitor pool is featured by the expression of the transcription factor TCF-1 and it is comprised by the stem-like precursor and the precursor subsets. The stem-like precursor pool is tissue resident and expresses the chemokine receptor CXCR5. CD8^+^ CXCR5^+^ T cells strongly infiltrate HCC, and their infiltration has been suggested to predict a better prognosis linked to HCC cell death [78,79]. CXCR5 also mediates homing of B cells [80], which are also enriched in HCC and interacts with T cells, enhancing immune activation, and this finding is linked to better outcome [81]. Therefore, tertiary lymphoid structures enriched in CXCR5+ B cells and TCF1+ stem-like CD8^+^ T cells correlates with responses to immunotherapy [82]. The stem-like precursor pool differentiates into the precursor pool that is also TCF-1 positive and can dedifferentiate into the stem-like precursor but also give rise to the effector progeny. These precursors pools have self-renewal potential and a catabolic mitochondrial metabolism, based on fatty acid oxidation and oxidative phosphorylation which promotes long-survival of these T cells [83,84]. They express memory markers such as IL-7 receptor (CD127), and lower levels of PD-1. It is noteworthy to know that these PD-1^dim^ precursor pool is the subset sensitive to the immunoregulatory strategies based on PD-1 blockade [85]. During persistent antigen stimulation, this subset develops an epigenetic signature to adapt to this high pressive antigenic landscape [50,51]. Although PD-1 based therapy should target the progenitor pool, this approach should be combined with the modulation of epigenetic changes [86] to improve current anti-PD-1/PD-L1 immunotherapies, since PD-1 blocking alone does not remodel the exhausted epigenetic profile [49], which would make temporary the response to anti-PD-1 treatment [60]. In this sense, γc cytokines could play a role in epigenetic and metabolic remodeling that should be explored [74,87].

## 5. PD-1 Mechanisms Involved in CD8 T Cell Exhaustion

ICI therapy has signified an outstanding breakthrough in Oncology in the last decade, with objective response rates of 15–20% in more than 10 different kind of tumors [88,89]. Blocking PD-1/PD-L1 pathway is the prototype of this therapy and it is the backbone of different combination therapies including other IC, cytotoxic chemotherapy, ablatives strategies, or molecular targeted therapy [13]. Recently, the mechanism of action of this inhibitory IC has been described [90,91]. PD-1 is a member of the CD28 family that, unlike the other CD28 family members, generates negative signals after TCR triggering [92]. PD-1 negatively signals by preferentially dephosphorylating CD28 by PD-1 recruited Shp2 phosphatases. In fact, CD28 expression is necessary for the recovery of T cells subjected to anti-PD-1 immunotherapy. PD-1/PD-L1 interaction dephosphorylates CD28 but not TCR. Therefore, the TCR triggering impairment observed in exhausted T cells during persistent antigenic stimulation is due to the loss of the positive CD28 co-stimulation. CD28 is a positive co-stimulatory signal involved in naïve and memory T cell activation upon TCR triggering. The PD-1^dim^ stem-like precursor T cells express CD28, making these cells sensitive to PD-1 modulation while the generated progeny lacks the expression of this receptor, making less likely the response to PD-1 blocking [93]. In a cancer animal model, PD1-1 blockade did not restore anti-tumor T cell function in presence of CD28 blocking antibodies, linked to a tumor growth similar to the not-treated control group [90]. Besides this CD28 dependent PD-1 inhibitory mechanism, it has been recently reported that PD-1 can also inhibit T cell activation upon TCR triggering in absence of CD28 co-stimulation [94]. Therefore, PD-1 blocking could be also useful as a short-term strategy to rescue CD28 negative effector cells but, to obtain a more persistent response, it will probably be necessary to restore the functionality of CD28 positive memory T cells. All these data explain the basic mechanisms involved in the impairment of the reactivity in the exhausted T cell response after T cell triggering. Nevertheless, the exhaustion profile is linked to other T cell changes that should also be rescued in addition to restoring the TCR signaling [95]. This issue renders the possibility of searching for synergic treatments in combination with anti-PD-1 therapy to increase the OS in HCC.

## 6. Results of PD-1 Blocking Based Combination Therapy on CD8^+^ Cells in HCC

To personalize the immunotherapy, it is necessary to know the quality of the immune response that is fighting against HCC. As discussed in Section 2, around 10–20% of HCC has reactive CD8 T cells prone to respond to ICI (IFNγ dominant cluster). Another 40% displays an inflammatory immunotolerant phenotype (Wound healing and inflammatory clusters) featured by an imbalance between pro-tumoral and antitumoral responses (Figure 1). These cases could need other treatments besides ICI to boost immune response such as anti-angiogenics or TGF-β inhibitors. Finally, around 40–45% of cases are not immunoactive (Lymphocyte depleted cluster). This last group could need synergic strategies to favor intra-tumoral trafficking of CD8^+^ T cells. To reach this goal, ablation therapy, chemo/radioembolization, and chemotherapy are being tested to increase antigen exposure, and WNT inhibitors besides ICI [17,18], because these cold tumors are characterized by *CTNNB1* and *AXIN1* mutations in the WNT-β catenin pathway [32]. All these strategies are currently being explored, but in the future a more specific approach should be tried to focus these actions on the CD8^+^ progenitor pool to make the response sustainable. Consequently, these approaches should be also combined with therapies involved in dealing with the metabolic impairments linked to the exhaustion imprints of the progenitor tumor specific CD8^+^ T cells [96,97]. Finally, anti-PD-1/PD-L1 treatment can be applied simultaneously with these combinations to fulfil these objectives but can also be clinically useful as neoadjuvant therapy to increase the response to systemic therapies [98].

### 6.1. PD-1/PD-L1 Blockade Monotherapy

Nivolumab and pembrolizumab are the first humanized monoclonal antibodies (mAb) against PD-1 tested in HCC patients [99]. The initial data on objective response rate (ORR) and median OS for both nivolumab and pembrolizumab in patients previously treated with sorafenib were around 15% and 1 year, respectively, in the initial clinical trials (CheckMate 040 and KEYNOTE-224) [28,29]. This response rate would correspond with the reported frequency of cases with IFNγ dominant HCC cluster [17]. Nevertheless, subsequent phase 3 clinical trials, nivolumab vs. sorafenib as first line (CheckMate 459) and pembrolizumab vs. placebo as second line (KEYNOTE-240) did not demonstrate improvement in OS [100]. This lack of benefit could be due to the selection of patients that could include more cases with non-immune-type HCC but also could be the result of tumor resistance to PD-1 treatment in the initial responders after repetitive cycles of treatment. Although anti-PD-1/PD-L1 therapy changes the tumor microenvironment by increasing the infiltrating CD8 T cells with cytotoxic activity [30,101] and can improve CD8 T cell primming by dendritic cells [102], these effects could be temporary due to the induction of compensatory inhibitory pathways [60]. Other anti-PD-1/PD-L1 monotherapies as second-line after sorafenib failure have shown also promising results, such as durvalumab (anti-PD-L1) and cemiplimab (anti-PD-1) [18]. These PD-1/PD-L1 monotherapy protocols could be effective in those HCC case with an IFN-γ dominant CD8 T cell infiltrate but those cases without an immune-class phenotype could also benefit from ICI after additional synergic therapies. Moreover, in the initial responders the role of epigenetic imprints in the exhausted T cells should be considered, which could lead to the induction of compensatory inhibitory mechanisms and resistance during long-lasting therapy. Therefore, to increase the number of responders and to avoid the development of resistance to treatment, several combinatory strategies with synergic actions are currently underway.

### 6.2. PD-1/PD-L1 Blockade Plus Antiangiogenics

The angiogenesis inhibitors are the current basic treatment in advanced HCC [103]. Multi-kinase inhibitors (MKIs) may modulate antitumor immunity through both angiogenesis-dependent and independent mechanisms (Table 1). Potentially beneficial effects on anti-tumor immunity may result from increased M1 polarization of macrophages and stimulation of CD8 T cell function. Nevertheless, high dosage of the MKI may contribute to immune suppression in the tumor microenvironment [104].

The wound healing HCC type is very rich in the expression of angiogenic genes, which could be a rationale to use anti-angiogenics in combination with ICI to change the balance between CD8 response and HCC in this type of tumor. The tumor hypoxic environment promotes high VEGF intra-tumoral level, which induces immunosuppressive effects by primming negative immunomodulatory inflammatory cells and by impairing the maturation of dendritic cells [105,106]. Clinical trials combining these drugs with anti-PD-1/PD-L1 are ongoing [18]. VEGF inhibition or blocking its receptors increases intra-tumoral infiltration and survival of cytotoxic T lymphocytes and decreases regulatory T lymphocyte recruitment, resulting in a more favorable immune microenvironment for ICI antitumoral activity [107,108]. These data highlight the important correlation of HCC neo-angiogenesis and the induction of immunosuppression produced by the hypoxic intra-tumoral environment. Levatinib is a MKI targeting VEGFR 1–3, fibroblast growth factor receptor 1–4, platelet derived growth factor receptor, RET and KIT with similar anti-tumor activity to sorafenib [103]. In an HCC animal model, levatinib shows an immunomodulatory activity featured by increasing CD8^+^ cell tumor infiltration and by decreasing intra-tumoral M2 TAM population [109]. This effect leads to a synergy with nivolumab treatment, enhancing tumor regression, which was attenuated by CD8^+^ T cell depletion, highlighting the role of CD8^+^ T cells in the response to the combination levatinib plus nivolumab [110]. Levatinib plus pembrolizumab combination have been also been tested in a phase I clinical trial with tumor reduction in most patients [111] and, according to this encouraging result, a new phase 3 clinical trial is underway, comparing levatinib vs. levatinib plus pembrolizumab in naïve patients (NCT03713593). This positive synergic effect has also been described combining durvalumab (anti-PD-L1) with ramucirumab (anti-VEGFR 2), which displayed a particularly high anti-tumor response in patients with high PD-L1 expression [112]. In the same line, the IMbrave 150 trial has tested the combination of atezolimumab (anti-PD-L1) plus bevacizumab (anti-VEGF mAb) with one third of cases with radiological response that were maintained by most of the responders during follow up [113]. Cabozantinib is another MKI of c-Met and VEGFR-2, and also inhibits AXL and RET [103]. The treatment with cabozantinib increases the expression of HLA-I molecules on tumor cells, leading to a greater sensitivity to killing by specific CD8 T cells. Besides this effect, cabozantinib also decreases the tumor infiltration by Tregs and MDSCs and increases the frequency of M1 TAM, which all together will improve the CD8^+^ T cell response [13]. Sorafenib is the first MKI clinically used in HCC that acts on RAF and VEGFR-2 and -3. This drug in a low dose promotes the intra-tumoral CD8^+^ T cell response and increases the M1 polarization of TAM, but in a high dose sorafenib displays the opposite effect [104,114,115]. The dose-effect on T cell response with MKI must be addressed in the future to obtain the best synergy with anti-PD-1 treatment. The potential immunological effects of MKI are summarized in Table 1. To sum up, based on all these clinical and experimental data, several clinical trials testing the combination of antiangiogenic plus PD-1/PD-L1 inhibitor are underway [116]. These combinations yield synergic effect and they could be specially interesting in “Immune class” tumors without reactive IFN-γ secreting CD8 T cells (wound healing and inflammatory clusters).

### 6.3. PD-1/PD-L1 Blockade Plus Anti-TGF-β1

TGF-β1 promotes tumor cell proliferation and invasion [117] and attenuates tumor response to PD-L1 blockade by excluding T cells [118]. In half of the lymphocyte rich tumors with exhausted T cells, the TFG-β1 induced genes are up-regulated [18,32]. Although the TGF-β1 dominant cluster is poorly represented in HCC [17], TGF-β1 plays a role in HCC pathogenesis. Galunisertib is TGF-β1R inhibitor that is been tested as a single drug and in combination with sorafenib in a phase I-II clinical trial in HCC [119,120]. This molecule has also tested in combination with stereotactic body radiotherapy to promote T-cell antitumor immunity, showing an anti-HCC effect featured by an increase in the CD8 PD1+ TIGIT+ population [121]. This drug could be an attractive strategy to increase the effect of PD-1/PD-L1 blockade on exhausted T cells after showing ability in increasing intratumoral infiltration of PD-1-expressing cytotoxic T cells. In fact, fusion proteins targeting both TGF-β1 and anti-PD-L1 are underway [122].

### 6.4. PD-1/PD-L1 Blockade Plus Other Immune Checkpoint Modulators

HCC-specific CD8 T cells can express other inhibitory checkpoints that are also up-regulated as exhaustion compensatory mechanism after anti-PD-1/PD-L1 treatment. Therefore, strategies to combined blockade of different ICs could be effective to rescue these cells from exhaustion but increasing the risk of adverse events such as autoimmune phenomenon. CTLA-4 competes with more avidity than CD28 for the same ligands CD80 and CD86, transmitting an inhibitory signal to the cell, instead of the positive co-stimulation mediated by CD28. CTLA-4 is highly expressed by tumor infiltrating HCC-specific CD8 T cells and the level of expression inversely correlates with the effector function. The CTLA-4 blockade with blocking anti-CTLA4 mAb improved CD8 T cell reactivity [16]. Based on these translational data, CTLA-4 blockade with tremilimumab has been tested in clinical trials as monotherapy or combined with ablation therapy in pre-treated patients with an elevated disease control rate [123,124]. Another anti-CTLA-4 inhibitory mAb, ipilumumab, has been tested in the phase II CheckMate040 clinical trial in sorafenib pre-treated patients to enhance the response against PD-1/PD-L1 blockade with a median OS of 24 months and with a 34% objective response rate and 5% cases with complete response, but has increased the rate of adverse events with respect to the arms without anti-CTLA-4 [125].

The effect of in-vitro blocking of other inhibitory ICs on HCC-specific CD8 T cells have already been tested with encouraging results. Blocking mAbs against TIM3 and LAG3 in association with anti-PD-L1 restore HCC-specific CD8 T cell response with synergic effect [21,53,121]. Nevertheless, these combinations have not been assayed in-vivo. Another strategy to boost HCC-specific CD8 T cells could be to associate the stimulation of positive co-stimulatory receptors, such as 4-1BB. In this line, a fusion protein containing PD1 and 4-1BBL has shown to block the PD-1/PD-L1 interaction and to trigger 4-1BB in HCC-specific CD8^+^ T cells in an HCC animal model, enhancing antitumor immune response and long-lasting tumor regression [71]. This same strategy has been applied in a transgenic HCC mouse model combining anti-PD-1 blockade with the synergic stimulation of two positive co-stimulatory molecules (4-1BB and tumor necrosis factor receptor superfamily, member 4 (OX40)). The triple combination extended the survival of mice bearing hepatocellular carcinoma in a CD8-dependent fashion [126].

### 6.5. PD-1/PD-L1 Blockade Plus Strategies to Increase HCC Antigen Presentation

In those lymphocyte depleted HCCs, strategies to release neo-antigens could have a positive effect in the induction of an HCC-specific CD8 T cell response. To achieve this goal, systemic chemotherapy, thermal ablation, or chemoembolization could be used. In an animal HCC model the combination of microwave ablation plus anti-PD-1/anti-CTLA-4 enhanced intra-tumoral infiltration of cytotoxic T lymphocytes and extended survival [127]. Although chemotherapy agents are generally considered immunosuppressive, these treatments through tumor cell death promotes the releasing of HCC neo-antigens that could let the priming of HCC-specific CD8 T cell response. The combination of cytotoxic agents (FOLFOX4 or GEMOX) plus anti-PD-1 (camrelizumab) in naïve HCC patients showed an overall rate response of 26.5% [128]. Currently, clinical trials to check anti-PD1 plus chemoembolization (NCT03572582), radioembolization (NCT03033446, NCT03099564), and stereotactic body radiation therapy (NCT03316872) are underway.

### 6.6. PD-1/PD-L1 Blockade Plus γc Cytokines

Along with these synergic strategies to increase the number of sensitive cases and to enhance the response to ICI, other approaches must be considered to direct these therapies to improve the HCC-specific progenitor pool. The addition of survival cytokines involved in memory differentiation and with the ability of metabolism reprogramming [129] would favor a healthier precursor pool to sustain a long-lasting efficient effector subset, able to keep HCC under control. IL-15 and IL-21 have shown promising in vitro results, favoring the development of HCC-specific stem-cell memory and central memory cells [130]. Additionally, IL-15 in a HCC animal model promotes tumor control, based on the long-term expansion of tumor-specific CD8 T cells, increasing IFNγ secretion and cytotoxicity, and also decreasing the expression of co-inhibitory molecules on dendritic cells [131]. Intra-tumoral IL-15 can also enable T cells to eliminate cancer cells lacking cognate antigens, which could be especially interesting in those lymphocyte-depleted HCCs [132]. Moreover, those HCCs with IL-21-secreting CD8 CXCR5 T cells develop a better outcome than cases without this type of cell [78]. The combination of PD-1/PD-L1 blockade with these γ-chain cytokines involved in precursor CD8 T cell generation [133] has not yet been explored in HCC but it has shown promising results in a cancer animal model [134].

### 6.7. PD-1/PD-L1 Blockade Plus Other Immunotherapies

Adoptive T cell transfer (ACT) therapy of TCR-engineered T cells [135,136], chimeric antigen receptor T cells (CAR-T) [137], cytokine-induced killer cells (CIK) [138], or ex-vivo restored T cells [35] could benefit from combined anti-PD-1 treatment to sustain the response of the transferred anti-tumoral T cells. The anti-HCC vaccines, both peptide- and dendritic cell-based vaccines, could also benefit from anti-PD-1 combined therapy [139]. Finally, the oncolytic virotherapy could also improves its results by adding anti-PD-1 treatment. This therapy is a novel approach based in the lyse of tumor cells by a vaccina virus that promotes the release of tumor antigens that will trigger an anti-tumoral immune response, which could be resistant to exhaustion by blocking the PD-1/PD-L1 pathway [140].

## 7. Conclusions

The antigen-specific CD8 T cell response against HCC-neoantigens carries out a key role in HCC control. Nevertheless, these cells become exhausted, being unable to keep tumor cells under control. These exhausted cells are featured by the expression of several negative regulatory checkpoints. PD-1 is the prototype of negative immune checkpoint and its blockade has been shown to rescue T cells from exhaustion in chronic viral infections and tumors. The HCC-specific CD8 T cells can be either exhausted in different grade or being inexistent according to the HCC immune class. The IFN-γ dominant HCC cluster displays exhausted T cells prone to respond to single PD-1/PD-L1 blockade. The immunotolerant types (wound healing and inflammatory clusters) will need the combination of PD-1/PD-L1 blocking plus other strategies to change the balance between pro-tumoral and anti-tumoral responses. The combination with anti-angiogenic drugs has shown promising results in this setting. Finally, the lymphocyte depleted tumors could need the association of lytic treatments to increase the exposure to neo-antigens to induce an HCC-specific CD8^+^ T cell response. To get this goal, the use of ablative treatments or chemotherapy could have an important role. Besides these strategies, new approaches focused on directing these treatments to improve the precursor pool of HCC-specific CD8 T cells could make long-lasting the response to these PD-1 based combination therapies. The progressive better understanding of the immune tumoral microenvironment and the mechanisms involved in the CD8 T cell exhaustion will probably permit soon to improve the overall survival of advanced HCC patients. Probably, PD-1/PD-L1 blockade-based combination therapies will be the backbone of these strategies to increase duration of response and number of responders.

## Figures and Tables

**Figure 1 cancers-13-01922-f001:**
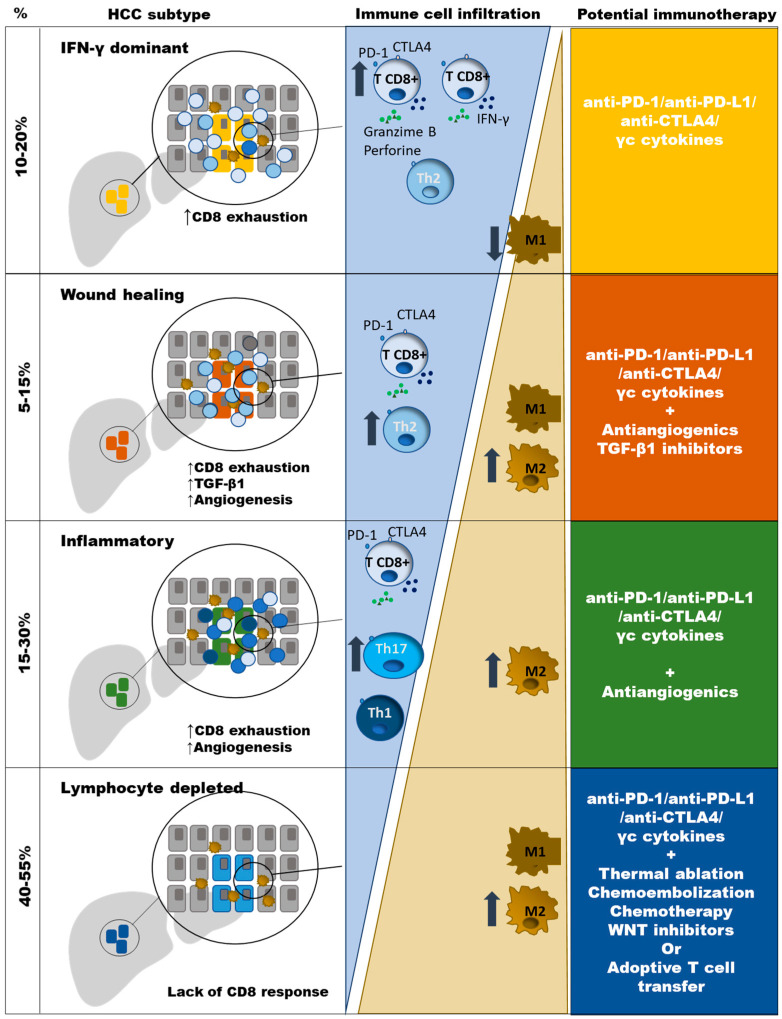
Type of anti-PD-1/PD-L1 based combination therapy, according to immune HCC subtypes. Progressive impairment in the balance between CD8 T cells and M2 TAM is observed in different HCC clusters. Neo-angiogenesis is induced in “Wound healing” and “Inflammatory” types. TGF-b1 genes are up-regulated in “Wound healing” HCC type. Absence of CD8 T cells in the “Lymphocyte depleted” type could be due to the lack of neoantigens or T cell deletion. TGF-β1 dominant and the immunologically quiet clusters are poorly represented in HCC.

**Figure 2 cancers-13-01922-f002:**
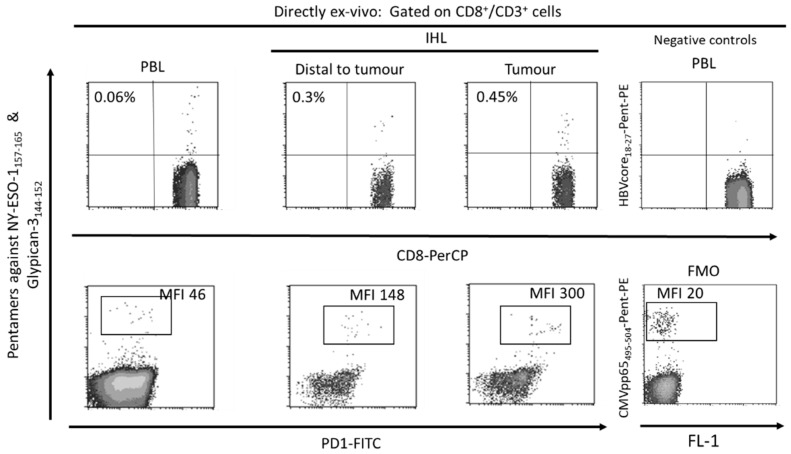
Flow-cytometric analysis of peripheral blood and intrahepatic lymphocytes in a patient with a Barcelona Clinic Liver Cancer Stage B hepatocellular carcinoma with hepatitis C cirrhosis, HBV negative, CMV positive. The ex-vivo CD8^+^ T cell response against the HLA-A2 restricted HCC epitopes Glypican-3_144–152_ and NY-ESO-1_157–165_ was tested by staining with CD8, CD3 and HLA-I pentameric complexes loaded with specific peptides. A tumoral sequestration of HCC-specific CD8 T cells and a PD-1 up-regulation gradient between peripheral blood and tumor was observed. PBL: peripheral blood lymphocytes, IHL: intrahepatic lymphocytes, MFI: mean fluorescence intensity, FMO: fluorescence minus one.

**Figure 3 cancers-13-01922-f003:**
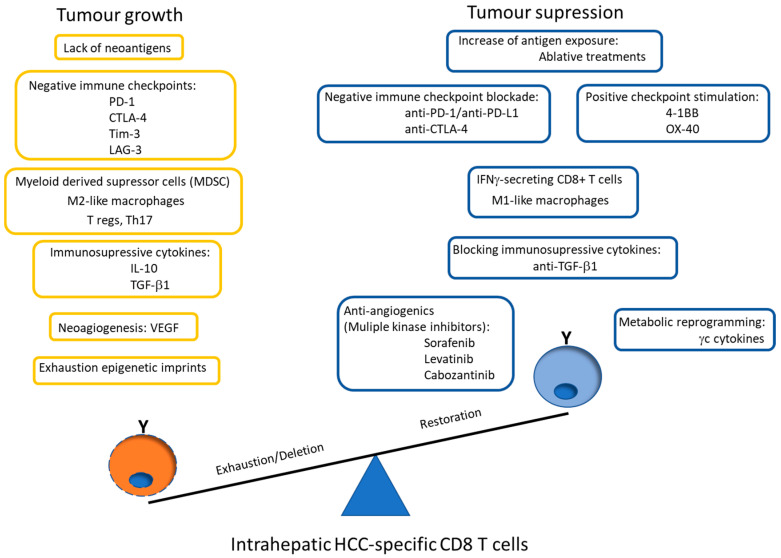
Scheme showing the potential mechanisms involved in HCC-specific CD8 T cell impairment. The graph also highlights the possible PD-1/PD-L1 blockade-based combination therapies to rescue effector and precursor HCC-specific cytotoxic T cell response. PD-1: programmed cell death protein 1, CTLA-4: cytotoxic T lymphocyte antigen-4, Tim-3: T cell immunoglobulin domain and mucin domain, LAG-3: lymphocyte-activation gene 3, T regs: CD4 T regulatory cells, Th: T helper, IL: interleukin, TFG: tumor growth factor, VEGF: vascular endothelial growth factor, 4-1BB: tumor necrosis factor receptor superfamily member 9, OX-40: Tumor necrosis factor receptor superfamily member 4, γc: gamma-chain.

**Table 1 cancers-13-01922-t001:** Immunological effects of multi tyrosine kinase inhibitors in hepatocellular carcinoma [13]. TAM: tumor associated macrophages, MDSC: myeloid derived suppressor cells, Treg: CD4 T regulatory cell, DC: dendritic cell, NK: natural killer cell.

Multi Tyrosine Kinase Inhibitor	M1 TAM	M2 TAM	MDSC	CD8	Treg	DC	NK
Sorafenib (low dose)	↑↑		↓	↑↑	↓	↑	↑
Regorafenib	↑	↓					
Cabozantinib	↑		↓	↑	↓		
Lenvatinib		↓		↑			

↑: moderate increase, ↓: decrease, ↑↑: high increase.

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
