# Peer review of "Anti-PD-1/PD-L1 Based Combination Immunotherapy to Boost Antigen-Specific CD8+ T Cell Response in Hepatocellular Carcinoma"

_cancers, 2021, doi:10.3390/cancers13081922_

Round 1

Reviewer 1 Report

The article by Peña-Asensio et al is a review addressing the issue of immunotherapy of hepatocellular carcinomas (HCC). The subject is of great interest taking into account the recent development of successful treatments of cancers based on Immune Checkpoint Inhibitors (ICI) blockades and the lack of efficient clinical options in HCC.

- The review is well documented presenting a large overview on the current knowledge of immune responses in the liver and HCC based on large number of citations.

- The heterogeneity of HCC is presented in details with the focus on immune parameters of the tumor, its environment in the liver. Several HCC types are defined accordingly to their immunogenic features.

- A critical analysis of predicting parameters and of the efficacy of treatment are presented in association with the immune status.

- Combinations of therapies based on actions modulating CD8+ T cells are discussed for each type of HCC.

Altogether, the review provides a large amount of updated information bridging insight in immune mechanisms with immunotherapies of HCC, emphasizing the diversity of liver cancers and the complexity of the immune functions of the liver, which needs to conciliate the tolerance and the responses to the exposures to antigens.

Specific comments

1- Figure 1: some texts in the center panel are too small, thus should be larger (name of markers and cytokines); the 3 panels should be defined by a title on the top; these suggested modifications would improve the clarity of the figure.

2- Figure 2: the flow cytometry has to show control of the specificity of HLA-I pentamer_ NY-ESO1 & Gpc3 peptide labelling of T cells. The number of positive cells is low, as expected, therefore any background would impact the data. For instance, the same labelling on a non-HCC donor.

3- Figure 3: some errors must be corrected:

”Mieloid derived suppressor …” by “Myeloid …”

“IFNg-secreting CD8+ …” by “ “IFNg-secreting   “ (symbol)

“Metabolic reprogramming: gc cytokines …” by “Metabolic reprogramming: gc cytokines …” (symbol)

4- The authors did not discuss the etiology of HCC, and how it affects the HCC types in terms of immune status. As the text is already long, I would recommend adding some information in section 2 about viral, non-viral (metabolic, toxic) etiology in the different types of HCC.

5- Lines 123-124: provide a reference to these data

Author Response

Specific comments

1- Figure 1: some texts in the centre panel are too small, thus should be larger (name of markers and cytokines); the 3 panels should be defined by a title on the top; these suggested modifications would improve the clarity of the figure.

Response: the figure 1 has been modified according to reviewer’s suggestion.

2- Figure 2: the flow cytometry has to show control of the specificity of HLA-I pentamer_ NY-ESO1 & Gpc3 peptide labelling of T cells. The number of positive cells is low, as expected, therefore any background would impact the data. For instance, the same labelling on a non-HCC donor.

Response: the figure 2 has been modified according to reviewer’s suggestion

3- Figure 3: some errors must be corrected:

”Mieloid derived suppressor …” by “Myeloid …”

“IFNg-secreting CD8+ …” by “ “IFNg-secreting   “ (symbol)

“Metabolic reprogramming: gc cytokines …” by “Metabolic reprogramming: gc cytokines …” (symbol)

Response: the misspelling mistakes in figure 3 has been corrected according to reviewer’s suggestion

4- The authors did not discuss the aetiology of HCC, and how it affects the HCC types in terms of immune status. As the text is already long, I would recommend adding some information in section 2 about viral, non-viral (metabolic, toxic) aetiology in the different types of HCC.

Response: a new paragraph has been added in section 2, enumerating the different common HCC aetiologies and a new reference that describes the immune infiltrates in viral and non-viral HCC has been also added. Also, a potential weaker HCC-specific CD8 T cell response in non-alcoholic steatohepatitis related HCC respect to viral-associated HCC is also highlighted, which could impact in immunotherapy (Page 2, Lines 74-82).

5- Lines 123-124: provide a reference to these data

Response: the reference to the statement referred by the reviwer has been added to the text (Page 4, line 143).

Reviewer 2 Report

This review summarized the HCC tumor immunotherapy based on the immune cell profile difference or PD-1 expression. the author introduced HCC subtype based on the immune cell profiling or PD-1 expression and TIL positive or negative or based on the expression profiling. Then the author further summarized the CD8+T cell relationship with HCC treatment and the epigenetic changes in exhausted CD8+ T tells, connection of macrophages with CD8+ T cells. Then the author discussed the stem-like precursor pool and how could target them, and the PD-1 molecular mechanisms and potential outcomes for anti-PD-1 based combination therapy.

In summary, this review provided substantial information about currently progression of immunotherapy in HCC treatments. However, the writing and organization needs major improvement. It is extremely difficult to follow through each topic.

Major points”

  1. The structure of this review should be reorganized. For example, the therapy and the classification can be divided into two different topics instead of blend in together. the infiltration of M2 and M1 macrophages and their relationship with CD8 + cells can be one topic.
  2. The author should try to reorganize the review based on the organizations of figures. The author tried to put every information in this review, making the review very difficult to follow. Please sort the information and follow your title to summarize the information.
  3. About figures and Tables:
    1. Maybe the author should make a figure that summarize all possible tumor types based on immune cell population or PD-1 expression or gene expression profiling. which can substitute the figure 1
    2. Figure 1. the title of three column needs to be added in the figure. like type of tumor, immune cell type and possible therapy; the percentage information can be added to figure 1 as well
    3. Did Figure 2 made by the author or adopted from a published paper? if it is adopted from other paper, it needs citation information. The author cited figure 2 in line 245 and 246, but I cann’t tell how the sentences related to figure 2.
    4. Figure 3 needs more detailed explanation in the legend.
    5. Table 1 needs citation
  4. Possible therapy combination discussion should get to the point. If the combination has not been investigated yet, the author can skip it or put in a session for future possible combined therapy

Minor points:

  1. The abstract needs improvement: less background but more about what exactly has been discussed in this review.
  2. Rewirte the sentence in line 55 and 56“To boost and to extend the effect of this approach, it is being explored the synergy of this therapy with other strategies acting to different levels, such as blocking other negative IC,” make it shorter and be readable. There are many more sentences needs rewrite.
  3. in Line 67 to 70 ” Immunotolerant …The immunogenic tumours comprises those cases that are interferon (IFN)-gama_dominant” the classification stated here is not consistent with the figure 1. make sure the content is consistent with the figures. it will be helpful to put the percentage of the tumor type in the figure 1 as well
  4. what does “HCC control” refers to in line 61? maybe use a less confusing wording here. like HCC treatment? Control sometimes refers to an experiment group without any treatment.
  5. what does “HLA” stand for? full name needs to be given before abbreviation.

Author Response

Major points

  1. The structure of this review should be reorganized. For example, the therapy and the classification can be divided into two different topics instead of blend in together. The infiltration of M2 and M1 macrophages and their relationship with CD8 + cells can be one topic.

Response: we disagree with the reviewer. The section 2 of the review is only focused on the classification of different HCC tumours according to immune response. The information given about some clinical trails is to support some statements of the section, but we do not try to make a review of the clinical results of anti-PD1 treatments in this work. We think that are currently many papers dealing with this issue. We neither think that macrophages/CD8 cells should need a different section. It is true that high densities of myeloid cells, that is, macrophages and myeloid derived suppressor cells (MDSC), correlate with poor prognosis. When it has been characterized, it appears that the negatively impacting macrophages are of the M2 phenotype. In any case, the correlation between macrophage density and patient survival is less significant than that of T cells, particularly CD8+ T cells. In a review about general immunopathogenesis of HCC this issue could be a separate topic but, in our review, it should be discussed in the same section about the interactions modulating CD8 T cell response. In fact, the information about macrophages is already discussed in section 3 (Page 6, lines 233-244).

  1. The author should try to reorganize the review based on the organizations of figures. The author tried to put every information in this review, making the review very difficult to follow. Please sort the information and follow your title to summarize the information.

Response: we think that we follow a clear flow in our review. In the introduction we describe the objective of our review. In the section 2 we summarize the different rationales to subdivide the HCC types according to the immune response in order to make the reader able to understand the potential PD-1 based immunotherapeutic strategies that will be discussed at the end of the review. Also, we try to give some data about chance to respond to anti-PD-1 treatment based on immunological and tumour variables. In this section we show figure 1 to summarize one of the models of HCC immune subtypes currently published, but in this section, we discuss two other possible models. In section 3 we present the key cell type of this review, which is the exhausted HCC-specific CD8 T cells and we describe the interaction with the HCC immune landscape. In this section we show one experiment from our own lab to illustrate the intrahepatic sequestration and PD-1 expression of HCC-specific CD8 T cells. Section 4 and 5 discuss about the target CD8 subpopulation for anti-PD1 treatment and how PD-1 up-regulation impairs CD8 T cell reactivity. Finally, after giving to the reader the basic information to understand why anti-PD-1 treatment could be an effective strategy to restore HCC-specific CD8 T cell response, we show the issues of isolated PD-1 treatment and discuss the potential improvement by adding other therapies. Therefore, we think the structure of the manuscript should not be modified in order to preserve the rationale design by the authors.

  1. About figures and Tables:
  1. Maybe the author should make a figure that summarize all possible tumour types based on immune cell population or PD-1 expression or gene expression profiling, which can substitute the figure 1

Response: we have followed the reviewer’s advice in the point 3.2 and we have modified the figure 2, according to reviewer’s suggestion, instead of making a new one.

  1. Figure 1. the title of three column needs to be added in the figure. like type of tumour, immune cell type and possible therapy; the percentage information can be added to figure 1 as well

Response: according to reviewer’s suggestion the titles on the different columns of figure one have been added and the percentage of occurrence of the different HCC subtypes have also been stated in a new column.

  1. Did Figure 2 made by the author or adopted from a published paper? if it is adopted from other paper, it needs citation information. The author cited figure 2 in line 245 and 246, but I can’t tell how the sentences related to figure 2.

Response: this figure 2 belongs to our research group and it has not been published yet. This information has been added into the figure’s legend. The figure 2 does not relate to the final sentences of the paragraph but to the whole section, to show the tumoral sequestration and PD-1 up-regulation of HCC-specific CD8 T cells. To clarify this, the cite of this figure has been moved to sentences clearly related with this figure in this section (Page 5, lines 183 and 190).

  1. Figure 3 needs more detailed explanation in the legend.

Response: the meanings of the acronyms used in the figure have been added to make the figure more self-explicative.

  1. Table 1 needs citation

Response: The table is cited in the line 384 and a reference has been added to the table legend to attest the information stated by the table.

  1. Possible therapy combination discussion should get to the point. If the combination has not been investigated yet, the author can skip it or put in a session for future possible combined therapy

Response: we think that discussing future possible anti-PD-1-based combination therapies enriches the topic of the results of PD-1 blocking combination therapy on CD8+T cells.  This item does not try to discuss only the results shown by the clinical trials, but it pretends to display a wide translational view of all tested treatment combinations with potential effect on HCC-specific CD8 T cells. This information can be interesting for the general reader since usually these data can be found only in journals of immunology and our approach can open new insights in Oncologists to other combination therapies than those with anti-angiogenics or anti-CTLA4. We think that in the treatment section we clearly state when the treatment has been tested in human clinical trials in HCC or it has been tested in HCC animal models, cell lines or in other cancers.

Minor points:

  1. The abstract needs improvement: less background but more about what exactly has been discussed in this review.

Response: the abstract has been modified enumerating the current PD-1 combination therapies that has been clinically tested and have shown effect on CD8 T cell response (Page 1, lines 43-44).

  1. Rewrite the sentence in line 55 and 56“To boost and to extend the effect of this approach, it is being explored the synergy of this therapy with other strategies acting to different levels, such as blocking other negative IC,” make it shorter and be readable. There are many more sentences needs rewrite.

Response: This sentence is long because it enumerates different potential PD-1 based combinatory therapies. We have rewrite it, citing between brackets the potential treatments to make it easier to read (Page 2, lines 65-69).

  1. In line 67 to 70 ” Immunotolerant …The immunogenic tumours comprises those cases that are interferon (IFN)-gama_dominant” the classification stated here is not consistent with the figure 1. make sure the content is consistent with the figures. it will be helpful to put the percentage of the tumor type in the figure 1 as well

Response: The classification stated in the section is consistent with the figure 1, but as stated in lines 88-89, the TGFb1 dominant and the immunologically quiet clusters are poorly represented in HCC and for this reason those types are not shown in the figure 1. We have added this information to the figure legend to make it clear (Page 4, lines 109-110).

  1. What does “HCC control” refers to in line 61? maybe use a less confusing wording here. like HCC treatment? Control sometimes refers to an experiment group without any treatment.

Response: According to reviewer’s suggestion we have changed “control” by “treatment” (Page 2, lines 70-71) 

  1. What does “HLA” stand for? full name needs to be given before abbreviation.

Response: we have added the meaning of the HLA acronym (human leukocyte antigen) before using it (Page 5, line 196).